# Packaging Materials Based on Styrene-Isoprene-Styrene Triblock Copolymer Modified with Graphene

**DOI:** 10.3390/polym15020353

**Published:** 2023-01-09

**Authors:** Traian Zaharescu, Cristina Banciu

**Affiliations:** INCDIE ICPE CA, 313 Splaiul Unirii, 030138 Bucharest, Romania

**Keywords:** styrene-isoprene-styrene (SIS), graphene, rosemary, durability analysis, thermal degradation, stabilization, chemiluminescence

## Abstract

This study presents the improved stabilization effects of graphene on a polymer substrate, namely a styrene-isoprene-styrene triblock copolymer (SIS) which creates opportunities for long-term applications and radiation processing. The added graphene has a remarkable activity on the protection of polymer against their oxidation due to the penetration of free macroradical fragments into the free interlayer space. The chemiluminescence procedure used for the evaluation of the progress of oxidation reveals the delaying effect of oxidative degradation by the doubling extension of oxidation induction time, when the material formulation containing graphene is oxidized at 130 °C. The pristine polymer that is thermally aged requires an activation energy of 142 kJ mol^−1^, while the modified material needs 148, 158 and 169 kJ mol^−1^, for the oxidative degradation in the presence of 1, 2 and, respectively, 3 wt% of graphene. The contribution of graphene content (1 wt%) on the stability improvement of SIS is demonstrated by the increase of onset oxidation temperature from 190 °C for neat polymer to 196 °C in the presence of graphene and to 205 °C for the polymer stabilized with graphene and rosemary extract. The addition of graphene into the polymer formulations is a successful method for enlarging durability instead of the modification of receipt with synthesis antioxidants. The presumable applications of these studied materials cover the areas of medical wear, food packaging, commodities, sealing gaskets and others that may also be included through the products for nuclear power plants.

## 1. Introduction

The duration of any material is directly related to its oxidation strength that is the main characteristic feature taken into account for the material delivery [1]. The prevention of oxidation is the most functional effect that certifies the stabilization activity offered by the involved additive. The large variety of the compounds possessing the inhibition ability upon polymer degradation, namely synthesis antioxidants such as hindered phenols [2,3], natural extracts [4,5], inorganic structures as oxides [6], polyhedral oligomeric silsesquioxane (POSS) [7,8], complexes [9] or multicomponent blends [10] radiation crosslinking [11,12] have been investigated. The problem regarding the improvement of the polymer stability in the presence of antioxidants appears as the background requirement for new applications involving the long-life operation under severe conditions [13,14], the recycling [15] or food handling [16,17].

The preparation of graphene structures [18,19,20] opens new ways for the improvement of functional properties in materials which are related to the penetration or retention of particles according with the application requirements: oil adsorption [21], humidity and temperature sensors [22], material crosslinking [23], and so on. The majority of applications concern the capacity of graphene structures for the scavenging molecules or their fragments in the free interlayer space. Only few reported papers deal with the antioxidant ability of graphene [24], but the information on polymer stabilization has not been published until now. This paper intends to fill the existing gap in this area of polymer stabilization.

The graphene/polymer composites may be a reliable solution for several materials with an improved lifetime [25]. These materials may have a wide variety of applications, because the inclusion of the ordinated carbon structure allows for the retention of scavenged degradation intermediates and the chain breaking in the ageing of polymers [26]. The concentration of classical antioxidants is usually around 0.5 wt% [27], but it may be adapted depending on its efficiency. The extension of stabilization activity of the engineering polymers would offer a suitable alternative application to any other efficient stabilizer, especially for insulated electrical cables. Because the production price of graphene at the lab scale is convenient, its stability becomes an important advantage [28]. Nowadays, the competition between several classes of stabilization reveals the unknown structures that are able to replace phenolic antioxidants, the classical protectors. An interesting structural modification of SIS molecules as an epoxidized form presents antioxidant features, whose stabilization activity was evaluated by DPPH testing [29].

This paper proposes an example through which the dangerous synthesis hindered phenols may be replaced with an inorganic structure, more stable and with similar efficiency. The present study analyses the optional but relevant material with which the extension of durability does not spoil the material compatibility with the human body. The reported results complete the information communicated in a previous paper, where graphene films incorporated in SIS minimize the oxidation of polymer phase in the solar cell applications [30]. The ideas concerning the control activity of graphene in polymerization [31] or the photoactuator function of SIS modified with oxidized graphene [32] may be considered as relevant aspects connecting the scavenging availability of graphene for the antioxidant activity.

The main desire involving polymer stabilization is related to the additive efficiency, along with the conservation or improvement in other material properties such as mechanical behavior [33]. The benefit of stabilization consists of the augmentation of service time, which diminishes the material and energy consumption and guarantees the safe handling and/or operation. Thus, the synthetic polymers whose lifetime is limited by their structures and environmental conditions require the mandatory addition of an appropriate oxidation protector, through which the delay of ageing is effectively provided. The graphene array may be considered as a suitable structure that provides the achievement of sharp stabilization at low concentrations [34].

Numerous application areas of polymer/graphene composites include this crystalline carbon configuration that involves the layer structures through which molecules or molecular fragments penetrate: the scaffolds based on the honeycomb structure of graphene are ideal providers of drugs in the reconstruction of bones, cartilages, and electroactive tissues [35]. A small group of examples includes thermal conductive composites including graphene nanoplates a good technological solution for the material application in construction [36]. The polymer materials containing graphene sheets are considered as excellent materials for antibacterial applications [37], while the detection of pressure changes by high sensitivity electronic measurements is possible by graphene-based piezoresistive sensors [38].

The transport processes achieved in the degrading polymers are associated with the diffusion of outer molecular oxygen and the conversion of oxygen-containing intermediates into stable products after the former structures migrate onto the reaction spots. It was previously demonstrated [39] that various forms of carbon may show the properties of convenient oxidation protectors. The ability of carbon materials to play the role of stabilizers is dependent on the free volume existing between the component particles and on the structuration of carbon atoms distribution.

The inclusion of various atoms other than carbon is described in [40], where the migration of oxygen is depicted as the mobile nanophase and the attachment is achieved by a multistage interaction process. The catching of foreign entities is obtained by the overlapping of electronic orbitals, which are free to be occupied in graphene.

Depending on the diffusion of ammonia into the graphene structure [41], the scavenging activity of graphene is effectively achieved by the interlayer adsorption and the electron availability of intermediate radicals. The covalent bonding of polymer/graphene composites [42] may be considered as a real approach of stabilization applications, when the scission of macromolecules is the source of reactive fragments. The competition between the two fate trends of radicals: oxidation and protection are the background of stability assays by which the graphene substrate shows its antioxidant features.

Unfortunately, there are no reports concerning the protection of graphene against the oxidation of polymers. The stabilization mechanism is quite different from the classical activity of phenolic antioxidants. Their protection action is based on the substitution of labile protons [43]. Because graphene is the thinnest known two-dimensional (2D) material, it is capable to act on both sides of the interlayer free volume [44], and the tightness becomes satisfactory for the locking oxidizable moieties. The stability of interlayer penetrating parts is assured by the Wan der Waals forces, and the sp^2^ hybridization maintains the integrity of graphene nanocomposites [45]. Starting from this background, the protection activity of graphene may be assessed as covalent functionalization.

The utilization of graphene polymer composites as drug delivery material [46], antifoaming polymer-modified cement composites [47], preparation of conductive materials [48,49], ecological treatment of water [50] and many others is based on the structural distributions of holes and electrons [51], whose densities are crucial elements for the scavenging efficiency and the gap depths determines the bonding strength. The graphene oxides as well as the reduced graphene oxides are preferentially used in the formulation of various complex structures such as smart hydrogels [52], protective films [53], nanofibers [54], nanobiosensors [55] or resistant scaffolds [56]. As the proof of stabilizing activity simultaneous with the effect of mechanical improvement, the processing effect on polypropylene was reported [57]. The barrier effect for the penetration of outer oxygen for feeding oxidation is also revealed for graphene nanocomposites [58].

This paper offers an efficient solution for the manufacture of several long-term products efficiently stabilized by an allotropic form of carbon, reduced graphene oxide (rGO), whose contribution to the improving durability is the basic feature for special applications such as electrical cable insulation and high performance capacitors, food packaging, commodities, sealing gaskets, sealants and gaskets destined for nuclear energetics, adhesive items, bags for the storage of medical wear and protective sheets for equipment with outdoor service. The reported results are possible due to the structural configuration of graphene that allows the penetration of free radicals in the material hexagonal network, where electronic cloud and oxygenated functions interact strongly with degradation intermediates, keeping them apart from the propagation stage of degradation. Thus, these multipurpose applications are possible due to the capacity of graphene to break the degradation chain by the scavenging of free molecular fragments that propagate oxidation during thermal ageing. This enumeration is a short list of products, where SIS improved with graphene powder may be used due to the functional properties available over an extended period of operation. The great advantages that recommend graphene as an antioxidant additive are: stability, efficiency, and comparable price with synthesis antioxidants.

## 2. Materials and Methods

### 2.1. Materials

The polymer material, styrene-isoprene-styrene triblock copolymer (SIS), was purchased from KRATON (Houston, TX, USA) as D1165 PT sort. This material was purchased as (1,7)-polyoxepan-2-one pellets with an average diameter of ~3 mm. The styrene content is 30 wt%, density 1.145 g mL^−1^ @ 25 °C, average M_n_ 80,000 and the polydispersion index (M_w_/M_n_) is less than 2.

Graphene fraction was prepared in our laboratory as reduced graphite oxide (rGO). Natural graphite powder (flakes < 30 mm) was provided from KOH-I-NOOR Grafit (České Budějovice, Czech Republic). The p.a. reagents H_2_SO_4_ (96–98%), NaNO_3_ (99.5%), KMnO_4_ (99%), HCl (37%), H_2_O_2_ (30%) and ascorbic acid (99.6%) were purchased from Chemical Company (Iasi, Romania).

### 2.2. Preparation of Reduced Graphene Oxide

#### 2.2.1. Preparation of Graphene Oxide

The GO was synthesized from natural graphite powder by the modified Hummer’s method [59,60]. The natural graphite powder (1 g) was mixed on a magnetic stirrer along with 1 g NaNO_3_ and 100 mL H_2_SO_4_ keeping the temperature below 5 °C by immersion in an ice bath for 4 h. After homogenization under stirring (500 rpm), 6 g KMNO_4_ is added, keeping the temperature of the mixture below 10 °C for 1 h. The resulting solution was warmed to 35 °C and stirred for 1 h. Afterwards, the solution is diluted with 100 mL of distilled water, homogenized by stirring and heated to 95 °C and maintained at this temperature for 2 h. The solution was allowed to cool to room temperature, then 200 mL of distilled water was added and the mixture was stirred for 1 h. Forty milliliters of 30% H_2_O_2_ were added to complete the oxidation reaction, and the mixture was stirred 1 h at room temperature. Graphene oxide was formed, and the resulting slurry was washed with 5% HCl and distilled water to neutral pH. The water is completely removed from the resulting graphene oxide by freeze drying.

#### 2.2.2. Preparation of Reduced Graphene Oxide

The resulting GO (1 g) was diluted in 200 mL distilled water and sonicated at room temperature for 2 h to homogenously disperse the GO phase in water. Fifteen grams of ascorbic acid, used as reducing agent, was added to the resulting suspension and continuous stirring for 2 h at a temperature of 90–95 °C was applied. In the next step, the mixture was stirred for 20 h at room temperature during which time the reduced graphene oxide is formed. The suspension was washed with 5% HCl and distilled water to neutral pH. The water is completely removed from the resulting reduced graphene oxide by freeze drying.

#### 2.2.3. Structural Qualification of Reduced Graphene Oxide

The FTIR spectrum of prepared rGO is presented in Figure 1. The Fourier transform infrared (FTIR) spectroscopy was carried out using Bruker Tensor 27 IR (Natick, MA, USA). According with the spectral illustration, the content dominant concentration is alcoholic hydroxyl (3400 cm^−1^) [61]. The other two main peaks may be ascribed to carboxyl acid (1700 cm^−1^) [62] and 1100 cm^−1^ identified as C–O units [61]. The dissimilarity between the spectra of graphene oxide and reduced graphene oxide was reported [36]; it is relevant in the region around 3400 cm^−1^, where rGO presents a very prominent peak like our material.

### 2.3. Sample Processing

#### 2.3.1. Preparation

Small SIS balls were dissolved in chloroform by vigorous shaking until a clear liquid was obtained. From this mother solution, aliquots of 10 mL were transferred into small bottles, where appropriate rGO was added to obtain the concentrations of 1, 2 and 3 wt%. For the evaluation of coupling contribution of reduced graphene oxide (rGO) and rosemary extract (RM—processed in our laboratory [63]), a fourth solution was prepared. After their homogenization, some drops of each solution were poured in round aluminum trays (diameter 6 mm) where the solvent is removed by evaporation at room temperature. These films are obtained and further experimental activities go on like the procedure was previously presented [64].

#### 2.3.2. Thermal Treatment

The thermal treatment was carried out in a Lenton laboratory (Fairland, South Africa) oven with forced air convection. The samples were thermally treated in air at a constant temperature. The oven was stabilized at 80 °C, after which the samples were introduced and maintained at this temperature for 5, 10, 15 and 20 h. At the end of the exposure time, they were immediately removed from the oven and cooled to room temperature.

### 2.4. Measurements

Chemiluminescence (CL) determinations were achieved by means of a LUMIPOL 3 (Institute of Polymers, Slovak Academy, Bratislava, Slovak Republic) spectrometer, whose reading error in temperature values is ±0.5 °C. The proper parameters for the optimal experimental conditions were established: the heating rate of 10 °C min^−1^ for nonisothermal procedure was selected, while, for isothermal study, three temperatures (130 °C, 140 °C and 150 °C) allow the comparison of thermal stability and the calculation of activation energies required for the thermal degradation of the studied probes.

## 3. Results

The improvement of polymer stability requires an appropriate structure through which the free radicals are withdrawn from the degradation chain [65]. Graphene structures may be considered as the main component of nanomaterials [66], whose presence in the formulations brings about special features. However, the stability investigations were not reported, even though the spatial distribution of carbon layers allows the penetration of small particles [67]. The existence of some polar moieties, especially the hydroxyls [68], creates favorable conditions for an efficient interaction with free radicals that migrate through the graphene nanoplates by means of existing energetic gaps [69].

The thermal treatment of the polymer matrix reveals several basic aspects related to the fate of free radicals that result from the scission of SIS molecules. The most relevant feature that must be taken into account is the reduction of onset oxidation temperature (OOT), the kinetic parameter which indicates the start temperature of oxidation. If it is 208 °C for neat material, the preheated probes show lower temperatures around 192 °C (Figure 2). This behavior associated with the further fate of free radicals, which is determined by the competition between recombination and oxidation, influences the progress of thermal processing.

As it may be expected, the modification of thermal resistance due to the presence of radical scavengers, is suitably obtained. An important application of this benefit would be the recycling of plastics [70], when the “new” material is strengthened by the appropriate additive. The good oxidation resistances are confirmed by the CL nonisothermal investigations on a large temperature range extended to 200 °C (Figure 3). This feature is related to the efficient immobilization of radicals, even though they are at low concentrations (short thermal ageing period) or on longer heating ageing treatments. The differences between the oxidation processes appear at higher temperatures, unusual degradation conditions, when the local accumulation of free radicals determines the acceleration of material damaging. The presence of other components acting on the extension of thermal strength such as rosemary creates favorable opportunities for the improvement of durability for packaging materials.

The stabilization efficiency of the couple consisting of graphene (1 wt%) and rosemary (0.5 wt%) is revealed in Figure 4. The noticed gathering of the four curves indicates the prominent effect of the sustained heating for a minimum of 15 h. This time is sufficiently long for any deterioration on this material that happens during long-term service or the longer exposure to sun light. The cooperation between the two stabilizing structures is concerned for long accident times, when the degrading product is intimately protected.

The thermal processing of the SIS matrix reveals an interesting peculiarity related to the evolution of CL emission, which follows the classical mechanism based on the radical reactions [71]. The formation of free radicals as the first effect of energetic transfer provides reactive fragments that may react with diffused oxygen (oxidative degradation) or with each other (recombination). In the case of this polymer material, the oxidation is the first option over the period of induction and the start of propagation (Figure 5). While the number of scissions increases as the result of heating, the concentration of free radicals is enhanced and the competition between oxidation and crosslinking becomes more obvious. Because the oxidation is dependent on the diffusion rate of oxygen, it influences the proportion between the two main ways of radical decay. SIS shows a predominant oxidation for the first ten hours of heating (the samples processed at 5 h and 10 h) with the shortening of OIT. The extension of the treatment time induces an increase in the amounts of radicals. Due to the shortened distances between radicals, they may recombine with each other, and the CL emission intensity is diminished. Though the values of OIT are similar, about 150 min, the advance of degradation is obviously slowed down. This behavior is a specific feature for the materials where the recombination rate of radicals exceeds their oxidation.

The isothermal CL investigation highlights the effect of additive concentration on the relative durability of stabilized materials in comparison to pristine polymer (Figure 6).

The delay of oxidation is illustrated by the values of oxidation induction times (OIT) which increase as the amount of graphene is enhanced. For the SIS substrates that are not subjected to any environmental aggression, these figures are strikingly different than the value shown by the control specimen; they are minimum double and the protection activity determines an extension of the period during which the material remains undamaged. The radiation processing narrows the OIT values, but they are greater than the similar characteristic of pristine polymer. Another positive effect of the participation of graphene to the improvement of thermal stability is the significant extension of propagation stage of degradation, when the graphene nanoplates keep radicals tightly, and their further oxidation is blocked over a long time period. It can be significantly compared with the previous time, when the free radicals (the active intermediates that promote the progress of oxidation) are accumulated.

The failure of SIS products is directly dependent on several factors: the applied technology, the duration and intensity of energetic transfer, the material composition, especially the presence of stabilizing structures, the history of the item. The destructive actions of all these elements are minimized by the contribution of additives, whose efficiencies must be checked before their addition into new products.

The structural modifications during thermal degradation are revealed by the appropriate evaluation of stability, when the measured characteristics are correlated with the evolution of the state of the material. Chemiluminescence, an accurate analytical procedure, allows the counting of emitted photons by the conversion of carbonyl intermediates from the exited state into their background energetic level [72]. The scission of C–C σ bonds of tertiary carbon atoms where phenyl is placed, the breaking C=C bonds belonging to isoprene moieties, the removing of labile protons are the main ways through which the degradation starts in polymers [73]. The monitoring of photon emission during chemiluminescence measurements indicates the oxidation state of the probe and, simultaneously, the route of degradation by means of the comparative analysis of spectra.

The stabilization activity of graphene is sustained by the intercalation of radicals through the carbon layers. The electronic distribution over the planar carbon atoms allows the formation of enough strong bridges with the penetrated fragments. Thus, the delay of oxidation is achieved by the immobilization of degradation intermediates, which is proved by the extension of experimental times.

The reduced graphene oxide is a graphene where polar units containing oxygen are grafted on the constituent carbon atoms [74]. These units act additionally on the hydrocarbon fragments offering a supplementary contribution for the retention of radicals against their reaction with oxygen. It means that these materials are chain breaking structures. Similar terms are usually used for hindered phenolic antioxidants, but the action mechanisms are totally different.

The association of graphene with a natural antioxidant such as rosemary extract is an excellent version of stabilization formulation. In Table 1, the OIT values obtained by the protection of SIS against oxidative degradation by thermal processing are relevant for the selection of this couple for the manufacturing of ecological products such as packaging materials, whose thermal resistance must attain acceptable limits of durability.

The prediction of material durability based on the values of activation energy is a good method for the description of the competition between oxidation and temperature-dependent deterioration. When an investigated material presents a certain oxidation induction time (OIT), the effect of additives on the accumulation of oxygen-containing product is depicted by the duration of stabilized structures. The longer the steady state period on the start of oxidation, the more the participation of the protector delays the degradation (Figure 7). If the material presents a certain low consumption of oxygen due to the morphological characteristics, the action of antioxidant is confidently evaluated by activation energies.

The values of OOT shown in the last column of Table 1 are significant for the collaboration between the two manners of radical scavenging, which complete each other by the local blocking of oxidizing intermediate moieties. The increase in the temperature values characterizing the oxidation start is essential for all pre-heating times. It proves the stabilization on large areas of applications involving the thermal accidents or extended exposure to sun light.

The availability of reduced graphene oxide is illustrated by the values of activation energies that are involved in the delay of oxidative degradation, when the patterns contain increasing loadings of graphene additive (Table 2).

The protection activity is enhanced by the stacking effect [75], through which the graphene structure, coupled with the presence of polar oxygen containing functions, achieve the minimization of oxidation rate and the advanced deterioration of material quality. The antioxidant property of graphene oxide was demonstrated by its addition in food [76] or in the compatibilization of blood [77]. In fact, the stabilization efficiency of graphene oxide and, by extension, reduced graphene oxides, is promoted to a great measure by the functionalized materials with the oxygen containing units (hydroxyls and carbonyls) [24,78]. A special range in which graphene filler may be successfully is the polymer waste recycling [79]. This application is a proof of ability through which graphene acts by retention of oxygenated fragments or low weight molecules that appear during material ageing. A reverse situation, when graphene can be obtained from the processing of polymer wastes [80], represents a favorable opportunity for the conversion of pollutant material into useful additive that may be a helpful compound with special attractive properties for polymers.

In the action against oxidative degradation, the presence of several groups resulting from the oxidation of graphene [81] is associated with bending and stretching of interatomic bonds of carbon rings placed in the layer plans. These factors tightly keep the polymer fragments out from the degradation chains. The great challenge for the long-term stability of graphene composites is the limited amount of inorganic content because the segregation of nanoparticles diminishes the degree of freedom inside the polymer matrix [82,83]. The ability of graphene structures, especially the reduced graphene oxides to operate in aggressive conditions as the material formulation component of composing parts in electrical batteries [84,85,86], demonstrates that they are also appropriate for improvement of material stability and the contribution to the thermal stabilization may be suitably extended to any other economical areas that require the additional contribution to their stability. Fortunately, there are plenty of papers, where graphene structures form honeycomb configurations in inorganic systems [87,88], which may inspire more studies for finding appropriate solutions based on which the graphene based composite products can attain high performances and durabilities. Based on the mechanistic considerations on the activities of graphene structures, especially reduced graphene oxide, where oxygenated functions play the role of trapping agents, their polymer composites would act as suitable materials for a series of special applications such as membranes for fuel cells [89], cartilage tissue engineering [35,72], mechanically improved materials [90] and automotive items [91].

The rGO powder interfering in the mechanism of polymer degradation is an ecological component of products allowing long periods of operation because it is not damaged or consumed. Its inclusion as a stabilization component provides structural integrity to the material and the conservation of features.

## 4. Conclusions

The results presented in this paper prove protective action of reduced graphene oxide against oxidation by the penetration of polymer fragment into the free interlayer space. The presence of oxygen containing groups brings about supplementary stabilization strength. The optimization of stabilization efficiency is obtained by the suitable activity of graphene as the scavenger of degradation intermediates even during the operation on high temperature range exceeding 200 °C. The presence of graphene in the formulation of SIS based materials brings about a significant slowing down of the oxidation rate by the reduction of local concentration of free radicals, as well as the breaking degradation chain by the integration of degradation promoters into the lamellar configuration of additive. The increase in the graphene loading leads to the higher material stability extending the oxidation induction time from 130 min obtained for the ageing of pristine SIS at 130 °C to 380 min at the same temperature, when the oxidation takes place in the polymer containing 3 wt% graphene. The activation energies that increase as the rGO concentration become 1.2 times higher in the presence of 3 wt% graphene represent the reliable proofs for the selection of this additive as an efficient antioxidant for polymers. The polymer support, SIS copolymer, is an example for its class of materials, which may be improved when convenient loading of rGO is present. This study demonstrates that the classical antioxidants, hindered phenols or amine, may be replaced by this rGO due to the proper action as the chain breaker in the degradation mechanism. The availability for the material protection recommends it to be used in several applications in material manufacturing for food handling, medical wear, conductive sheets and bars, anticorrosive layers and commodities. Several applications of packaging materials, including food handling, are possible by the manufacturing of products based on the formulations that include a graphene/rosemary extract couple.

## Figures and Tables

**Figure 1 polymers-15-00353-f001:**
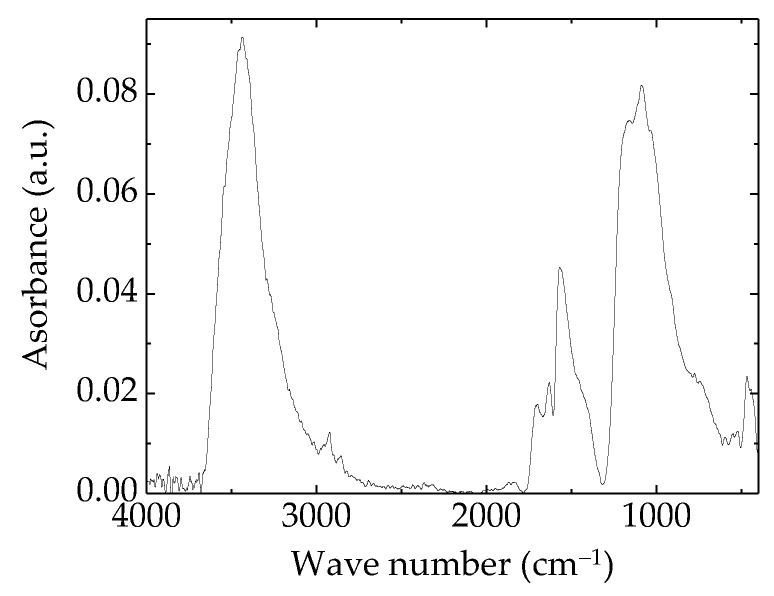
FTIR spectrum of prepared reduced graphene oxide.

**Figure 2 polymers-15-00353-f002:**
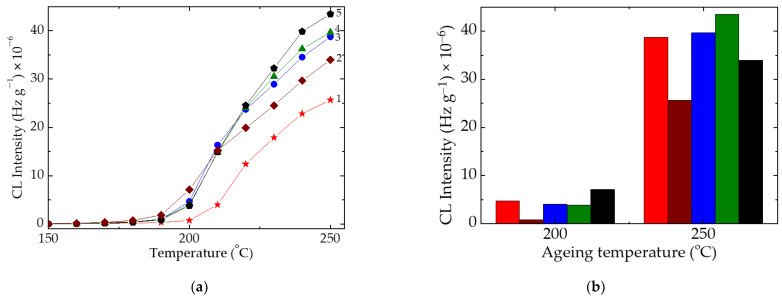
(**a**) Nonisothermal CL spectra recorded on pristine SIS samples subjected to a thermal ageing treatment at 80 °C at various heating times. (1) 0 h; (2) 5 h; (3) 10 h; (4) 15 h; (5) 20 h. Measurement heating rate: 10 °C·min^−1^; (**b**) histogram of CL intensity determined at two main temperatures (the color code is identical for the both figures).

**Figure 3 polymers-15-00353-f003:**
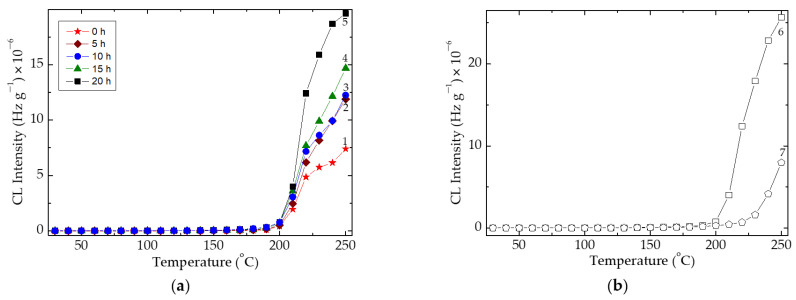
(**a**) Nonisothermal CL spectra recorded on SIS/graphene (1 wt%) samples after their thermal ageing treatment at 80 °C at various heating times. (1) 0 h; (2) 5 h; (3) 10 h; (4) 15 h; (5) 20 h; (**b**) nonisothermal CL spectra recorded on SIS (6) and SIS/graphene (1 wt%)/rosemary (0.5 wt%) (7) samples without pre-ageing. The heating rate of all measurement: 10 °C min^−1^.

**Figure 4 polymers-15-00353-f004:**
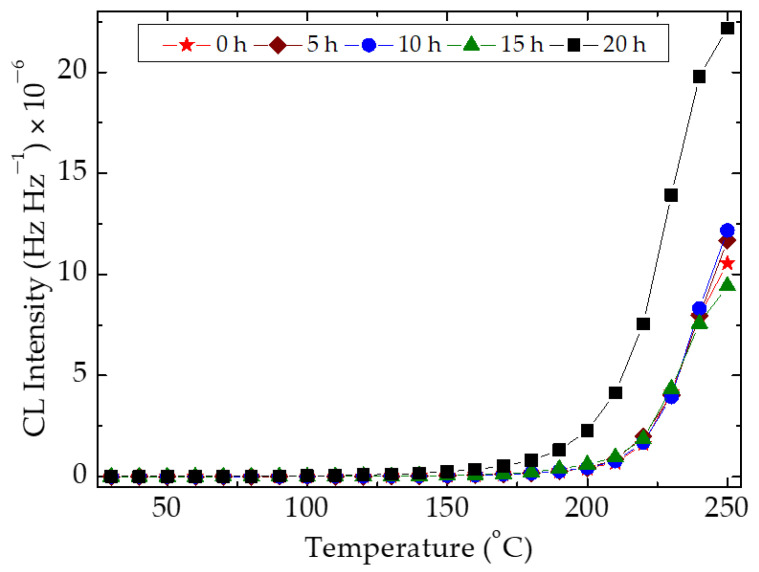
The nonisothermal CL curves recorded on the SIS samples improved by graphene (1 wt%)/rosemary (0.5 wt%) couple after their thermal ageing at 80 °C at various heating times. Heating rate: 10 °C min^−1^.

**Figure 5 polymers-15-00353-f005:**
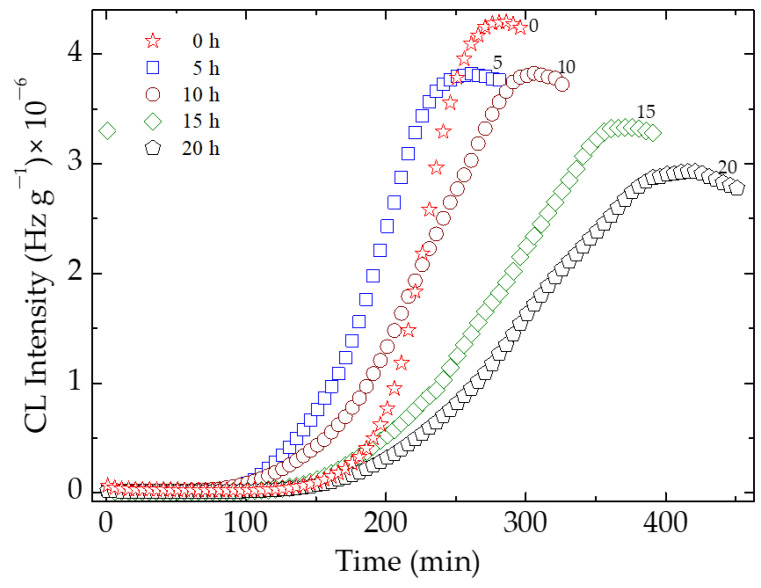
Isothermal CL spectra for neat SIS samples after their thermal treatment at 80 °C for various heating times. Heating temperature: 80 °C. Testing temperature: 130 °C. The mentioned figures denote the time of thermal treatment.

**Figure 6 polymers-15-00353-f006:**
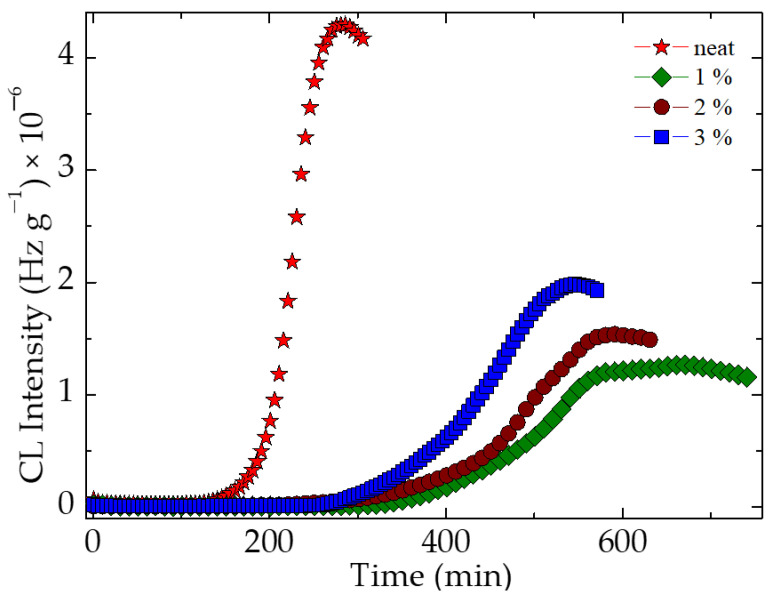
Isothermal CL spectra recorded on the samples of SIS containing different graphene loadings; measurement temperature: 130 °C.

**Figure 7 polymers-15-00353-f007:**
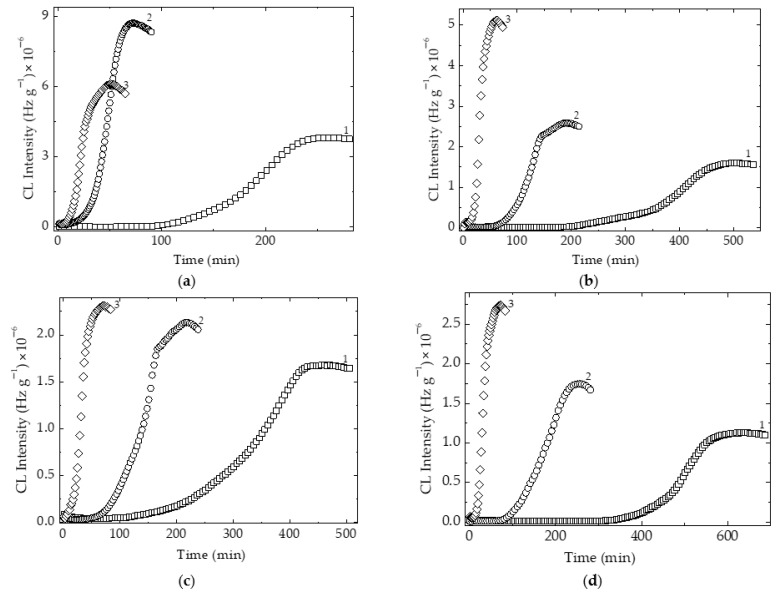
The isothermal CL spectra recorded on SIS samples modified by various graphene loadings. (**a**) free of additive; (**b**) graphene 1 wt%; (**c**) graphene 2 wt%; (**d**) graphene 3 wt%; Testing temperatures: (1) 130 °C; (2) 140 °C; (3) 150 °C.

**Table 1 polymers-15-00353-t001:** Onset oxidation temperatures for SIS samples modified with graphene (1 wt%) and rosemary extract (0.5 wt%).

Heating Time (h)	OOT (°C)		
	Pristine SIS	SIS + Graphene	SIS + Graphene + Rosemary Extract
0	202	215	219
5	201	207	220
10	198	201	219
15	196	198	215
20	190	196	205

**Table 2 polymers-15-00353-t002:** Activation energies required for the oxidation of SIS improved with various amounts of reduced graphene oxide.

Reduced Graphene Oxide Content (%)	OIT (min)			Correlation Factor	Activation Energy (kJ mol^−1^)
	130 °C	140 °C	150 °C		
0	130	43	18	0.99851	142
1	292	90	36	0.99833	148
2	185	98	31	0.99936	158
3	380	88	35	0.99325	169

## Data Availability

Not applicable.

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
