# Peer review of "Packaging Materials Based on Styrene-Isoprene-Styrene Triblock Copolymer Modified with Graphene"

_polymers, 2023, doi:10.3390/polym15020353_

Round 1

Reviewer 1 Report

Interesting results and novelty work. A paper focuses on Packaging materials based on styrene-isoprene-styrene triblock copolymer modified with graphene. Though the intention of the authors is highly commendable, there is lot of problems particularly in the presentation throughout the manuscript. Besides there are many grammatical mistakes throughout the manuscript, particularly in respect of use of singular and plural with the subject or verb. In view of the above comments, whole manuscript should be properly written to make it acceptable by Polymers journal. I highly recommended this article to be accepted and published in the revised version.

 Abstract:

The abstract given here starts without any background for the present work. Of course, it contains brief details about experimental aspects and the obtained results. However this abstract does not follow the norm of an abstract, which should state briefly:

1.     The purpose of the study undertaken, what are you trying to solve

2.     brief mention of experimental aspects (without using abbreviations)

3.     highlights of the results numerically

4.     Important conclusions based on the obtained results

5.     Potential applications

Therefore, it is suggested that the Abstract to be modified as per the suggestions given above.

 Introduction

Introduction section is long with a many references based on the literature survey conducted by the authors. This is very good. However, this lacks in proper presentation of literature survey, which should have been systematic whereby existing scientific gaps should have been brought out. This should have given justification for the present study, which should be followed by the objectives of this study. In fact there is large amount of literature available on the characterization of styrene-isoprene-styrene and its reinforcement. Similarly, a large number of methods to obtain these materials have been used mentioning their advantages and limitations. None of these have been brought out in this study whereby the authors have not justified why they have chosen the method they have used in their study.  It should be noted that normally 'Introduction' should give brief background through literature survey for the study citing previous published work where-by scientific gaps that exist should be brought out. This would have led to justification for the present study.  It is therefore suggested that ‘Introduction Section’ should be revised as suggested above because this Section is an important one from the point of view of taking up the present study.

Relevant article on reinforcement of polymer should be cited such:

Coatings 2021;11:1355. https://doi.org/10.3390/coatings11111355.

In introduction part authors mentioned about application.

What application is it? Packaging, construction, toys etc? Justified. Is there any evidence of the same? In my opinion the paper will have good merit if such applications can be demonstrated and reported. Can you give some example?

Please restate the objective of this paper.

Materials and Methods:

Normally, this section should have two main subsections. The first one is Materials which should give details of all materials used in the study, where from they were procured, known characteristics, if available (for e.g. where do you get it, what is the purity of the chemical and etc.). In this section you only mention 2 materials which are styrene-isoprene-styrene triblock copolymer (SIS) and styrene. Other materials used should also be mentioned. Graphene? Graphite oxide?

The second subsection should be Methods, where methodologies used in the study should be given in a systematic way using sub section with numbers for each of the properties. First the processing or preparation aspects of the final material should be given followed by the characterization of prepared materials including preparation of samples for any specific property or morphology studies should be presented in a systematic way. Here one should also clearly mention the number of samples used, any standards followed for variety of properties, make and model of the instruments used for characterization, their accuracy and experimental conditions used, etc.

It should be known to the authors when one publishes any scientific paper, the results presented therein should be such they should be reproducible by any other person when the experiment is repeated using the same materials. In the present paper, it would be difficult for any other person to repeat the experiments because the chosen materials do not have any pre-history, which is required for other researchers to carryout experiments to check the possible reproducibility of the procedure adopted by these authors.

Some of the paragraph should be under results and discussion and if it is already there then this becomes repetition and hence can be deleted. Secondly, this Section is methods and hence only results should be mentioned and then it should be discussed preferably comparing it with earlier reported similar results by other researchers.

Measurement should be elaborated more. 2.2.3. Chemiluminescence (CL), 2.2.4. XXXX, and others.

Results & Discussion

Well written and easy for the reader to understand what the authors have conveyed.

Some of the paragraph should be under Methods and if it is already there then this becomes repetition and hence can be deleted. Secondly, this Section is Results & Discussion and hence only results should be mentioned and then it should be discussed preferably comparing it with earlier reported similar results by other researchers.

Throughout the manuscript, there are no comparison had been done with other published journal. Therefore, please support your statements with other researcher’s work in the section result and discussion. It should be discussed preferably comparing it with earlier reported similar results by other researchers.

How many sample did for each experiment? Please do ANNOVA test and standard deviation for all data collected and presented.

Figure 2 b is not complete. What does it mean by red, brown, blue, green and black colour? Please do mention it. Same goes to Figure 2a and Figure 3b.

This section is not well written. Author should focused on the fundamental study. The current one is too short. Please elaborate.

Please combine Result and discussion part in one section.

Conclusions

Conclusions given here are do not reflect what had been achieved including many speculations. It is too long and should be in 1 paragraph. Hence these need to be suitably modified. It may be remembered that this Section forms a summary of all the major observations/ results obtained. Accordingly, here presentation should consist of the main Results or the observations of the study in short sentences probably with bullet points. This should stand alone or form a subsection of a Discussion or Results Section. Hence better to rewrite this Section based on the comments given in the whole text.

General Comments:

The paper though contains some interesting results and novelty work, it lacks in its proper presentation in the whole manuscript. Of course there is need for better language throughout the manuscript. It is suggested that the authors should take the help of native English speaking person to take care of this problem. In view of these, the paper is highly recommended and should be accepted for publication in the revised form. It is suggested that the authors should revise the paper in the light of above comments/suggestions.

Author Response

Happy New Year !

Reviewer 2 Report

Dear authors,

The paper presents very important solution for various long-term products stabilized by an allotropic form of carbon, reduced graphene oxide. Its contribution to improving durability is the main feature for specific applications therefore it plays a very important role in many fields. The paper presents very important contribution to this field.

Therefore, it can be accepted in present form.

Author Response

The authors are doubtfully indebted to you for the appreciation of our manuscript.

Let wish you “Happy New Year!”

Prof. Traian Zaharescu

Dr. Cristina Banciu

Reviewer 3 Report

This study and the kind of experiments have the character of a 'model procedure' using a model substance (here KRATON D1165 PT Polymer). Also, the material was not extruded, with temperature impact, but solved, what reflects a procedure to produce a model sample what is a bit different from real life samples. That should be more clearly written.

L 33: Here it would be of interest if antioxidants are used in food packaging.

L 40: Are costs available?

L 41: What advantage. Pls. provide values.

L 49: How high is the concentration for antioxidants nowadays?

L 52: What means low and compared to what?

Introduction:

Pls. add more values. Presentation of the state of the art should be more quantitative.

Write intention of the study and hypothesis.

Information about the degradation of PS is fully missing and also how and if it is stabilized.

L 96: “KRATON® D1165 PT Polymer is used in pressure sensitive adhesives, hot melt spray, diaper adhesives, construction adhesives, coatings, and sealants.” [https://www.ulprospector.com/de/eu/Coatings/Detail/21707/560704/KRATON-D1165-PT-Polymer] The used material is for adhesives. That is not a packaging material as written in the title! Why was this polymer chosen? Why PS?

L 166-173: Belongs maybe better to introduction.

L 218: Oxidation can also cause crosslinking at some polymers. How to be sure here scission is taking place?

L 266-270: Is this an assumption or really proven?

L 280-281: This sentence seems ‘constructed’. The tested material is for hot melts not packaging materials.

Table 1: Only single measurements? Error values are missing.

L 298: ‘pre-hating’? Typo.

L 300: Sun light might cause other reaction paths, i.e., photo-oxidation. Normally, antioxidants do not have an impact on photo-oxidation but only autoxidation which can be induced by photo-oxidation.

L 301-311 / 320-324: Might be better shifted to introduction.

Table 2: Tg of SIS should be provided somewhere. (Around Tg and Tm activation energy changes.)

L 342-343: Is it an assumption or proven?

L 351: Here also costs are an issue and approval, e.g., as food contact material in packaging.

Author Response

Answers to the comments provided by referee #3’comments, suggestions and recommendations.

The authors thanks gratefully to you related to the spent effort for the quality improvement of our manuscript. It is a good chance to open the door of publication.

So, we answer to your comments as well as we can.

L 33: Here it would be of interest if antioxidants are used in food packaging.

The interest on the applications of antioxidants in food packaging is undoubtedly enormous; because of the handling safety of food is one of the essential conditions of health security. A plenty of versions are available in the literature and in the patent areas, but it is not a barrier for the new achievements, like our manuscript. In essence, the present manuscript offers a new alternative for the efficient stabilization of polymer, SIS being a simple example as polymer support.

L 40: Are costs available?

We believe that the addition of graphene into a polymer material requires a price similar with the synthesis antioxidants. However, it is not converted into quinone forms, like hindered phenols, which becomes poison in the hosting food.

L 41: What advantage. Pls. provide values.

The great advantages that recommend graphene as antioxidant additive are: stability, efficiency, comparable price (under our opinion) with synthesis antioxidants like IRGANOX series,

L 49: How high is the concentration for antioxidants nowadays?

The concentration of antioxidants in the polymer formulations depends strongly on the efficiency of additives.

L 52: What means low and compared to what?

We did not found these words. If you intend to highlight “Though the graphene additives are more or less expensive”, we changed this fragment with more clear other words: “Because the production price of graphene at the lab scale is convenient”

Introduction:

Pls. add more values. Presentation of the state of the art should be more quantitative.

We add some now information. Unfortunately, the stabilization activity of graphene in any structural presentation is about absent through the paper in this range. It is surprizing that this evident application did not gain the merited attention. This paper opens largely the application door for this purpose.

Write intention of the study and hypothesis.

We believe that we completed the Introduction accordingly.

Information about the degradation of PS is fully missing and also how and if it is stabilized.

Well, it is true. The degradation of styrene is different that it is happened in the present triblock polymer. Even though styrene presents a double bond, it disappears by polymerization with isoprene. More than that, the unsaturation presented in SIS belongs to isoprene moieties and the benzene rings do not contribute to the oxidative degradation. We believe that the degradation of SIS is not directly related to the reactivity of styrene. Sorry!

L 96: “KRATON® D1165 PT Polymer is used in pressure sensitive adhesives, hot melt spray, diaper adhesives, construction adhesives, coatings, and sealants.” [https://www.ulprospector.com/de/eu/Coatings/Detail/21707/560704/KRATON-D1165-PT-Polymer] The used material is for adhesives. That is not a packaging material as written in the title! Why was this polymer chosen? Why PS?

You are right. SIS is destined to the adhesive applications. But, if we make a parallel judgment with PP foils that is used as preservation sheets for food, we may extend the usage ranges of SIS onto the packaging material areas taking into consideration the adhesive behavior. If it would be associated with other less permeable material as the additional doubling sheet, we may accept SIS to gain this material quality.

We chose this polymer because it may be judged as a component layer in a multisheet product.

In our intention, there is not any connection between SIS and PS. They are totally different.

L 166-173: Belongs maybe better to introduction.

Please, accept the same place in the final version.

L 218: Oxidation can also cause crosslinking at some polymers. How to be sure here scission is taking place?

In the case of SIS, where the mechanistic approach considers the two divergent consumption ways: crosslinking and oxidation after the generation of free radicals by the scission of bonds, the crosslinking is the result of inactivation of radicals by protector.

L 266-270: Is this an assumption or really proven?

The intercalation of fragments is usually mentioned in the papers dealing with the activity of graphene, graphene oxide and reduced graphene oxide. It is neither my invention, nor my imagination. Beg your pardon!

L 280-281: This sentence seems ‘constructed’. The tested material is for hot melts not packaging materials.

Table 1: Only single measurements? Error values are missing.

The calculation of one vale of energy needs minimum three isotherms. If we should have an average value of Ea, we must accomplish 15 experimental records. For four graphene concentrations, minimum 60 CL measurement would be done, so, it means that the paper would be focused only on this purpose. However, the chemiluminescence is a very accurate procedure and it allows to consider the error in the evaluation of Ea as ± 1 kJ mol-1 taking into account our experience on more than 30 years of practice in CL studies. If you have a look on other our papers

Polymers 2022, 14(22), 4971; https://doi.org/10.3390/polym14224971

Polymers 2022, 14(21), 4737; https://doi.org/10.3390/polym14214737

Radiation Physics and Chemistry, 2022, 201, 110446

Radiation Physics and Chemistry, 2021, 183, 109401

and many others,

we never mentioned this evaluation error. All reviewers accepted this state.

L 298: ‘pre-hating’? Typo.

The typing mistake was solved.

L 300: Sun light might cause other reaction paths, i.e., photo-oxidation. Normally, antioxidants do not have an impact on photo-oxidation but only autoxidation which can be induced by photo-oxidation.

We insert our comment in the text for clarification of the involvement of graphene as antioxidant.

L 301-311 / 320-324: Might be better shifted to introduction.

We did it for the firstly mentioned lines. The second group remains on the same place, because we believe that it is more convenient. Sorry!

Table 2: Tg of SIS should be provided somewhere. (Around Tg and Tm activation energy changes.)

As the paper

Toporowski, P.M.; Roovers, J.E.L. Glass transition temperature of low molecular weight styrene–isoprene block copolymers. J. Polym. Sci., Polym. Chem. Ed. 1976, 14(9), 2233-2242. https://doi.org/10.1002/pol.1976.170140913

stated, the Tg values of SIS is placed around 100 oC, far from the temperature range (130-150 oC) of the determinations of activation energy.

L 342-343: Is it an assumption or proven?

It is an assumption sprung from our experience.

L 351: Here also costs are an issue and approval, e.g., as food contact material in packaging.

You are right; the appropriate testing costs.

The authors thank you again for the effort you spent for us and we consider your suggestions and comments as a helpful hand for the publication of our manuscript.

Let us wish you “Happy New Year!”

Prof. Traian Zaharescu

Dr. Cristina Banciu

Round 2

Reviewer 3 Report

Thank you for ammendments. In Fig. 1 is a typo.